# Generalized Adversarial Learning–An Innovative Unsupervised Paradigm In LLM's Calibration

## Abstract

Large-scale Question-Answering (QA) models have shown remarkable capabilities in various domains, but they often suffer from low confidence and reliability in their answers. To address this issue, most existing methods rely on supervised calibration, which requires labeled data and fine-tuning. However, these methods are costly, time-consuming, prone to overfitting, and lack interpretability and non-intrusiveness. In this paper, we propose a novel unsupervised learning paradigm called "Generalized Adversarial Learning" (GAL) to improve the calibration of large QA models. GAL views adversarial learning as a multi-agent game process, consisting of three components: the generator, the processor, and the inspector. The generator is the original large model that produces answers to questions. The inspector is a module that evaluates the answers and poses additional questions to probe the generator's knowledge. The inspector also updates a matrix that measures the confidence level of each answer and a vector that guides the processor. The processor is a module that adjusts the model weights using a convolutional kernel prediction method, which enables parallel processing of billions of parameters within acceptable time and memory cost. We explain the core principles and ideas behind GAL and present empirical evidence demonstrating its effectiveness, interpretability, and non-intrusive nature, achieving performance surpassing the state-of-the-art in some metrics even within the field of supervised learning.

## 1 Introduction

Question-Answering (QA) is a fundamental task in natural language processing (Lewis & Fan (2018)). With the development of large-scale pre-trained language models, such as GPT-4 (OpenAI (2023)) and Llama (Touvron et al. (2023)), QA models have achieved remarkable performance in various domains and applications. However, these models also have some limitations, such as generating incorrect or inconsistent answers (Azamfirei et al. (2023)), or being overconfident or underconfident in their answers. These limitations pose serious challenges for the trustworthiness and robustness of QA systems, especially in critical scenarios, such as medical diagnosis or legal advice. Therefore, it is essential to calibrate the QA models, that is, to align their confidence and accuracy, and to ensure that they can provide reliable and interpretable answers (Chen et al. (2022)).

Existing state-of-the-art calibration methods are mostly supervised methods refined from their unsupervised version and require labeled data or fine-tuning dataset (Wang et al. (2023)) , which are costly, time-consuming, and prone to overfitting. Moreover, they lack interpretability (Ye & Durrett (2021)) and non-intrusiveness, making them less suitable for practical applications. For example, learnable temperature scaling (Balanya et al. (2022)) requires a separate calibration dataset and cannot be learned during training.Learnable label smoothing (Chen et al. (2022)) introduces noise to the training labels, which may degrade the accuracy and confidence of the model.Learnable Ensembling involves multiple models, which increases the computational complexity and memory consumption . Unsupervised calibration methods, such as those based on hidden states or sample temperature, like label smoothing, deep ensembling (Lakshminarayanan et al. (2017b)) and data augmentation (Wei & Zou (2019)),are more easy-to-use and scalable, but they do not provide any feedback or explanation for the calibration process, nor do they allow for fine-grained control over the model's parameters.

In this paper, we propose a novel unsupervised learning paradigm called Generalized Adversarial Learning (GAL) to improve the calibration of LLMs without the need for supervision, while also enhancing their interpretability and non-intrusiveness. GAL views adversarial learning as a multi-agent game process, consisting of three components: the generator, the processor, and the inspector. The generator is the pre-trained LLM itself, which does not require any modification of model structure, and no extra train is needed. The inspector is a dynamic interactive module that poses additional questions based on the generator's answers and recursively updates a confidence matrix and a guidance vector. The processor is a module that directly operates on the model weights using a convolutional kernel prediction method, which enables parallel processing of billions of parameters within acceptable time and memory cost. The inspector and the processor cooperate to fine-tune the generator's weights in an unsupervised manner, while preserving its structure and function, and improving its calibration and reliability.

Our main contributions are as follows:

- We introduce a novel learning paradigm that separates the generator from the conventional adversarial framework and improves the model's interpretability, which avoids modifying or fine-tuning the generator, additional adversarial training or regularization, thus saving computation and complexity.

- We devise an unsupervised algorithm to enhance the confidence of large question-answering models, creating new opportunities for this field, which does not depend on labeled data or human evaluation, or limit by the domain or task, showing better generalization and scalability.

- We design a convolution-based method for weight-level fine-tuning, enabling smooth integration of the calibration process with the standard training process of large-scale models, which preserves the shape and structure of the weights, allowing low-invasive and parallel operations on a large number of parameters, thus increasing the efficiency and flexibility.

## 2 RELATED WORK

**Unsupervised Calibration Techniques** Recently, there has been a shift towards developing unsupervised calibration methods for LLMs (Lester et al. (2021)). Unlike traditional calibration techniques that often involve extensive model parameter modifications, unsupervised approaches utilize algorithms like temperature scaling (Guo et al. (2023)) and deep ensemble (Lakshminarayanan et al. (2017b)). These methods aspire to improve the performance of LLMs without resorting to supervised fine-tuning. However, they also introduce unique challenges that warrant further investigation (Bohdal et al. (2023)).The rise of Pre-trained Language Models (PLMs) has been accompanied by several calibration methodologies, such as temperature scaling (Kull et al. (2019)), label smoothing (Ghoshal et al. (2020)). Although promising, these methods are mostly unlearnable and hence leave certain facets of PLM calibration unaddressed (Jiang et al. (2021)).

**Calibration Enhancements via Collaborative Training** One significant advancement in the domain of LLM calibration is the collaborative training with various language model subsets (Chen et al. (2022)). This technique has showcased potential in elevating the calibration accuracy. Notably, recent studies have highlighted that the improvements brought about by this technique are robust and relatively unaffected by external variables (Wang et al. (2021)).Relying heavily on calibration datasets, they come with challenges such as high data collection costs. Furthermore, the need for extensive computational power and memory resources for fine-tuning large-scale models is another concern, especially in unsupervised setups.An important yet unresolved issue in the field is the interpretability gap: existing calibration techniques provide limited insights, thus potentially hindering their application in critical scenarios().

## 3 METHOD

In this section, we introduce the three components of GAL: the generator, the processor, and the inspector. We explain how they interact with each other to achieve unsupervised calibration of large QA models.

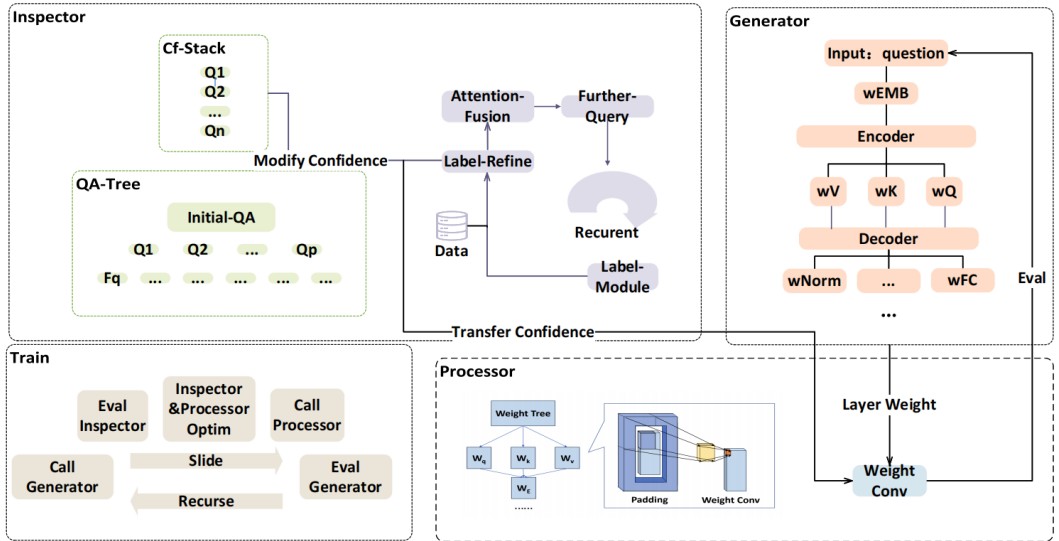

Figure 1: The main pipeline of calibration model using GAL paradigm. First, we evaluate the generator using data from the training set. Then, we pass the task and response to the inspector, which will generate an evaluation of the response and output guidance vectors to the processor. The processor will then optimize the generator by manipulating its weights. This approach overcomes the aforementioned challenges without altering the model structure and leveraging the existing training set, producing promising results. We refer to this paradigm as "Generalized Adversarial Learning".

### 3.1 PROCESSOR : WEIGHT CONVOLUTION MODEL

The role of our processor model is to optimize the model parameters based on the guidance vector generated by the inspector model. Considering that convolution is essentially the impulse response process, we implement this suggestion process by predicting the convolution kernel and convolving the weight matrix of model as shown in figure 2. To ensure that the shape of the weight matrix remains unchanged after convolution, we have designed padding, where the matrix is padded to make sure that the generator model continues to work normally after convolution. The specific formula is as follows:

$$P_i = \left\lfloor \frac{k_i - 1}{2} \right\rfloor, \quad \forall i \in \mathcal{K} \tag{1}$$

Consequently, the overall description of the entire weight convolution section can be formalized as follows:

The forward function is described as:

$$\mathcal{O}(k_i) = \mathbf{Conv}\left(\mathcal{D}(k_i), \mathrm{R}_{\mathrm{FC}(\mathcal{G}) \to 3 \times 3}(\mathcal{K}(\mathcal{G}))\right) \quad \forall k_i \in \mathcal{D} \tag{2}$$

Where $\mathcal{G}$ denote the guide vector, $\mathcal{D}$ represent the input dictionary, $\mathcal{O}$ be the output dictionary, $\mathcal{K}(\mathcal{G})$ be the convolution kernel generated from the guide vector, $\mathbf{Conv}(v, k)$ denote the convolution operation on input $v$ with kernel $k$, $\mathrm{R}_{a \times b \to m \times n}(\mathbf{x})$ denote the reshaping of vector $\mathbf{x}$ from dimensions $a \times b$ to $m \times n$.

### 3.2 INSPECTOR: RECURSIVE BAYESIAN APPROXIMATOR

Drawing inspiration from the human cognition process, we abstract the process of judging the confidence level of propositions into a system called Propositional Long-range Dependency System, and provide theoretical support in mathematics for studying this systemn, named as Recursive Bayesian Approximation.

We propose a system named proposition long-range dependency system to model the process of judging the confidence level of a proposition, as shown in figure 3. The system, denoted by $S$,

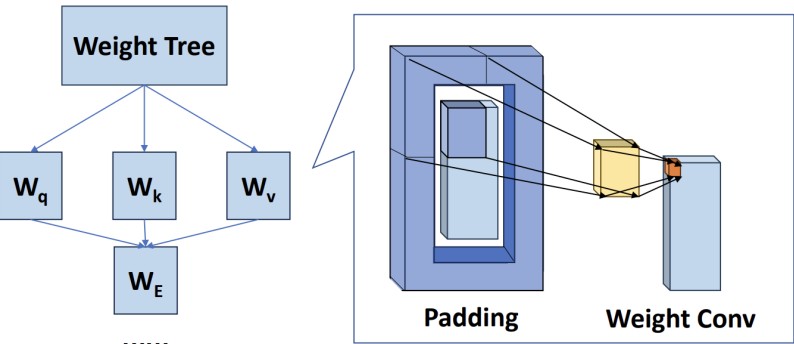

Figure 2: Weight conv. Through this design, adaptive updates for a large number of weights (on the order of magnitude of B) can be performed within an acceptable timeframe.

consists of $n$ sub-propositions $P_1, P_2, \ldots, P_n$, each with a confidence level $C_1, C_2, \ldots, C_n$, where $1 \leq i \leq n$. The system evolves over time according to a function $F$, which captures the mutual

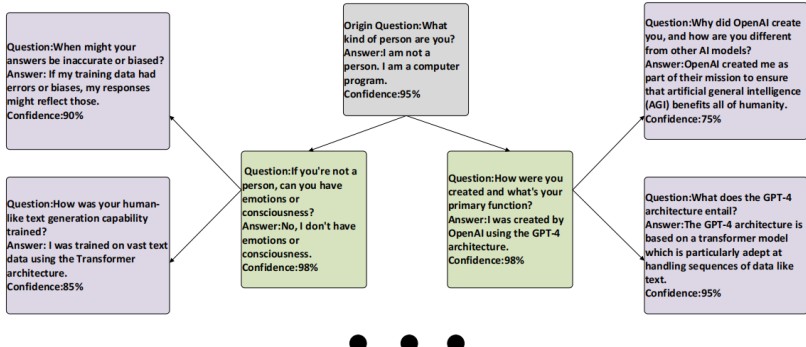

Figure 3: Example of tree-structural proposition and sub-proposition. The propositions and sub-propositions in the diagram are organized in a recursively constructed format, similar to the "follow-up questions" phenomenon in language. Each pair of question, answer, and confidence forms a node in the tree.

influence among the confidence levels of sub-propositions. The function $F$ can be written as a vector function, as follows:

$$\mathbf{C}^{(t+1)} = F(\mathbf{C}^{(t)}) \tag{3}$$

Here, $\mathbf{C}^{(t)}$ is the vector of confidence levels of all sub-propositions at time step $t$. This equation shows how the vector $\mathbf{C}^{(t+1)}$ is derived from the vector $\mathbf{C}^{(t)}$ through the function $F$, which describes the interactions between sub-propositions.

We assume that there are causal dependencies among the sub-propositions, meaning that a change in the confidence level of one sub-proposition may affect the confidence levels of others. These dependencies can be reflected by specific mathematical relationships within the function $F$, depending on the nature of the system and the details of the propositions.

The equation in line 8 represents the dynamic evolution of the system, describing how the confidence levels of sub-propositions change over time to reflect their mutual influence.

We consider a scenario where we have two hypotheses, $H$ and $H(n+1)$, with a causal relationship between them, and we want to update their relationship given some evidence. We can use conditional probabilities to express this relationship:

$$P(H|H(n+1), E) \text{ and } P(H(n+1)|H, E)$$

These represent how to update $H$ given $H(n+1)$ and $E$, and how to update $H(n+1)$ given $H$ and $E$, respectively.

Using Bayes' theorem, we can compute these conditional probabilities based on likelihood and prior probabilities. For example:

$$P(H|H(n+1), E) = \frac{P(H(n+1)|H, E) \cdot P(H|E)}{P(H(n+1)|E)} \tag{4}$$

$$P(H(n+1)|H, E) = \frac{P(H|H(n+1), E) \cdot P(H(n+1)|E)}{P(H|E)} \tag{5}$$

We can use these equations to update the relationship between $H$ and $H(n+1)$ given new evidence $E$. This can help us infer changes in their causal relationship to reflect new information.

We regard the process of judging the confidence level of a proposition as a system that evolves from the initial proposition. The evolutionary equation is shown in the algorithm 1. The inspector introduced below is an implementation of this theory.

We apply the above theory to the inspector. We first enrich the statement and the generator's answer with background information. Then, we use the label module to compute their confidence scores. We update the confidence matrix of all the remaining statements with each new score. Based on the current matrix and statement, we generate multiple questions with the question module and iterate the process for each question. Our aim is to approximate the true confidence level with external knowledge through effective and sufficient questioning.

---

**Algorithm 1** Propositional Long-range Dependency System Evolutionary Algorithm

1: **Input:** Initial truth value $P$, number of sub-propositions $M$, number of iterations $N$
2: **Output:** Final truth value $P_{\text{final}}$
3: $P_0 \leftarrow P$
4: **for** $i = 1$ to $M$ **do**
5:     $S_i \leftarrow f(P_i)$
6:     **for** $t = 1$ to $N$ **do**
7:         $P_i \leftarrow g(S_i)$
8:     **end for**
9: **end for**
10: $P_{\text{final}} \leftarrow h(P_1^{(N)}, P_2^{(N)}, \ldots, P_M^{(N)})$
11: **return** $P_{\text{final}}$

---

Where: $P$: Initial truth value of the proposition. $M$: Number of sub-propositions. $N$: Number of iterations. $P_i$: Truth value of the $i$-th sub-proposition. $S_i$: Score of the $i$-th sub-proposition. $f()$: Function that computes scores based on truth values. $g()$: Function that computes truth values based on scores. $h()$: Function that aggregates the truth values after $N$ iterations to produce the final score.

### 3.3 GENERATOR: PRE-TRAINED LLM

The generator is the large language model itself. It does not need any modifications to its structure or loss function, which is still cross-entropy. Unlike traditional adversarial generation methods, it interacts dynamically with the inspector and processor. The responses it generates are the basis for the next actions of the processor and inspector.

### 3.4 GAL MODEL TRAINING

For each training of the GAL model, the inspector is trained first. The generator can choose whether or not to update based on the training set. Then, the guidance vector output by the inspector is input to the processor, and the V-PPO algorithm is used for iterative modification and evaluation until the requirements are met.

For the training of the inspector, we applied the following regularization to ensure its good generalization.

$$R(\theta) = \lambda\|\theta\|_2^2 + \alpha D(\theta) + \beta N(\theta) \tag{6}$$

where: $R(\theta)$ is the total regularization term, $\lambda\|\theta\|_2^2$ represents the L2 regularization with $\lambda$ being the weight for this term, $D(\theta)$ denotes the dropout regularization term, weighted by $\alpha$, $N(\theta)$ represents the noise introduced during the label phase, weighted by $\beta$.

The training of GAL involves gradient descent for other parts, but we emphasize the training methods we designed for the processor part.

Validiation Proximal Policy Optimization (VPPO) is a reinforcement learning algorithm for the processor. It maximizes the performance of the processor based on proximal policy optimization (Schulman et al. (2017)). Here is the mathematical representation of the VPPO algorithm:

The objective is to adjust the model's weights to optimize its task performance. The state $s_t$ is the current weights, and the action $a_t$ is the weight changes. The reward $r_t$ is the model's performance, usually as the negative error. The model's state is its weights, and the action is a slight modification to them. The reward function $r(s, a)$ is the negative error after weight adjustments, using a cross-entropy loss.

The VPPO objective function is:

$$L^{\text{clip}}(\theta) = \hat{\mathbb{E}}_t\left[\min\left(r_t(\theta)\hat{A}_t, \text{clip}(r_t(\theta), 1-\epsilon, 1+\epsilon)\hat{A}_t\right)\right] \tag{7}$$

where, - $r_t(\theta)$ is the ratio between the new and old policies, defined as $r_t(\theta) = \frac{\pi(a_t|s_t,\theta)}{\pi_{\text{old}}(a_t|s_t)}$. - $\hat{A}_t$ is the normalized advantage function, representing the advantage of taking action $a_t$ in state $s_t$ relative to the average action. In the code, this is typically the reward the model gets for a given action minus the average reward.

The optimization goal of VPPO is to update the model's weights for better performance in the new state.

## 4 EXPERIMENTS AND RESULTS

We apply this method and mainstream methods in the training and fine-tuning processes of various datasets, testing and analyzing changes in their calibration. [1]

### 4.1 DO INSPECTOR WORKS?

To demonstrate the capability of the inspector, we tested and compared the metrics of various large language models and their Inspector Generator Score (Igs) on the test set. The results prove that the inspector possesses the ability to reflect quantity and trend in indicators such as confidence and can act as a supervisor within the model.

### 4.2 DO METHODS USED IN CALIBRATION USEFUL FOR QA?

We conducted comparisons across multiple methods on various datasets as shown in Table 2 and Table 3. . It can be observed that our model has achieved state-of-the-art (SOTA) performance in both supervised and unsupervised domain, and this method is even capable of elevating the performance level of vanilla models beyond that of some fine-tuned models.

### 4.3 TO HOW MUCH CAN WE EXPLAIN THIS METHOD?

Important intermediate quantities in this method, such as follow-up questions, confidence scores, and even the content of convolutional kernels, can all be intuitively observed and manipulated as shown in figure 5. Therefore, this provides our model with strong interpretability, which is undoubtedly crucial in the application domain of this model.

---

[1]For certain models that require excessively large training resources, we do not have sufficient computational power to conduct training tests.

| $Model$ | $Acc$ | $SConf$ | $Satisf$ | $Igs$ |
|---|---|---|---|---|
| $GPT2 - medium$ | 0.20 | 0.91 | 0.18 | 0.53 |
| $LLama2$ | 0.73 | 0.92 | 0.68 | 0.62 |
| $GPT3.5$ | 0.93 | 0.91 | 0.82 | 0.68 |
| $GPT4$ | 0.95 | 0.94 | 0.93 | 0.73 |

Table 1: Comparision between different metric on different models. The value of the Igs metric is positively correlated with other indicators.

| $Method$ | $Pre$ | $Squad^1$ | $DailyDialog^1$ | $TED^1$ |
|---|---|---|---|---|
| $Vanilla$ | 0.20 | 0.31 | 0.35 | 0.33 |
| $LS^2(Guo et al. (2017))$ | 0.18 | 0.33 | - | - |
| $EDA^2(Wei\&Zou (2019))$ | 0.26 | - | - | - |
| $Ensemble^2(Lakshminarayanan et al. (2017a))$ | 0.18 | 0.25 | 0.27 | 0.31 |
| $GAL^2$ | **0.36** | **0.36** | **0.42** | **0.35** |

Table 2: Comparision between different calibration methods on different datasets. the [1] means that the original model is fine-tuned on a dataset, the [2] means that it is unsupervised method. the [3] means that it is the SOTA of the supervised method.

| $Method$ | $Acc \uparrow$ | $ECE \downarrow$ | $Satisfy \uparrow$ | $BLEU \uparrow$ |
|---|---|---|---|---|
| $Vanilla$ | 0.20 | 0.921 | 0.18 | 0.018 |
| $LS^1(Guo et al. (2017))$ | 0.18 | 0.934 | 0.20 | 0.014 |
| $EDA^1(Wei\&Zou (2019))$ | 0.26 | 0.908 | **0.25** | **0.028** |
| $Ensemble^1(Lakshminarayanan et al. (2017a))$ | 0.18 | 0.916 | 0.20 | 0.021 |
| $E - MLP^2 Chen et al. (2022)$ | 0.22 | 0.905 | 0.21 | 0.024 |
| $DELP^2(Chen et al. (2022))$ | 0.27 | 0.863 | 0.23 | 0.022 |
| $TSLP^2(Chen et al. (2022))$ | 0.21 | 0.874 | 0.17 | 0.018 |
| $I - Iter^3(Chen et al. (2022))$ | 0.33 | 0.883 | 0.22 | 0.025 |
| $GAL^1$ | **0.36** | **0.834** | 0.22 | 0.020 |

Table 3: Comparision between different calibration methods. the [1] means that it is unsupervised method. The [2] means that it is supervised method. the [*] means that it is the SOTA of the supervised method. LS is label smooth, implemented in QA with the smooth of ids label in training period, EDA is a data augmentaion method that can be used to boost performance on training period. Ensemble is combination average score of 5 models

| Module | O1 | O2 | O3 |
|---|---|---|---|
| | Orig. mod. | FT with SQuAD | FT with SQuAD (sub.) + GAL |
| Dat | 0.20 | 0.31 | **0.36** |
| | Orig. mod. | Rem. Insp; rand. out. to Proc. guide vec. | Complete mod. |
| Insp | 0.20 | 0.28 | **0.36** |
| | Orig. mod. | Further-query with chatglm2-6b | Complete mod. with gpt3.5 |
| Fur | 0.20 | 0.28 | **0.36** |
| | Orig. mod. | Without Proc. | Complete mod. |
| Proc | 0.20 | 0.31 | **0.36** |
| | GPT2-med | GPT2-large | rwkv4-430m |
| Gen | 0.16 | 0.08 | 0.11 |

Table 4: Ablation study. From the accuracy results, it can be observed that both the GAL module and its internal design have an impact, and having these modules yields better results compared to fine-tuning with the dataset alone. Although the quality of the prompts can affect the model's performance, its reliance on prompting falls within an acceptable range (opting for models with over 100 billion parameters only yields a 0.05 improvement in accuracy over models with 6 billion parameters). Furthermore, experimental evidence with models of different architectures (RNN and Transformer) demonstrates that models of different architectures can benefit from this optimization method. Detailed settings can be seen in A.3

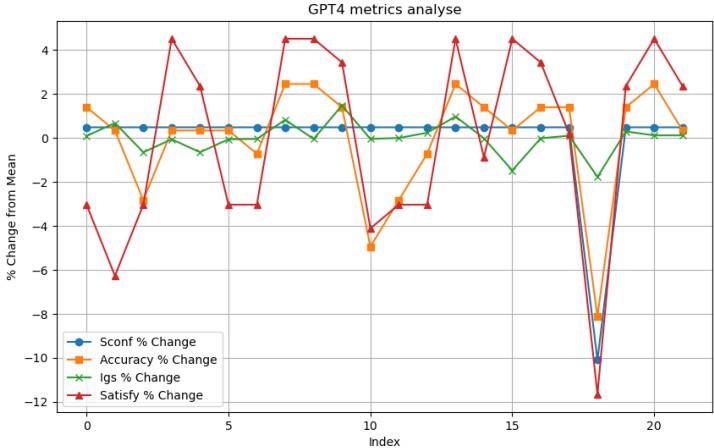

Figure 4: Evaluation and Trend Analysis of Various Metrics for the GPT-4 Model. From the graphs, it can be observed that there is a certain correlation between IGS and metrics led by accuracy (Acc), demonstrating the potential of IGS as an objective QA calibration metric to replace human intervention.

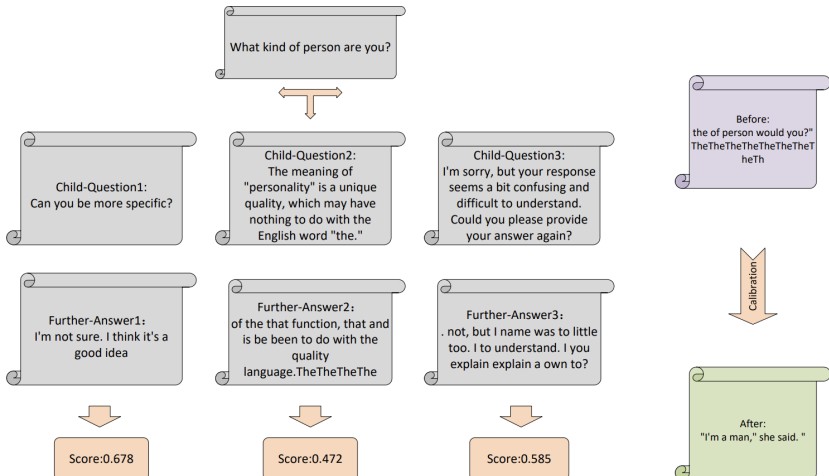

Figure 5: Visual workflow of our method. The intermediate results can be read, and the internals of the entire model are highly observable and controllable.

## 4.4 ABLATION STUDY

We conducted ablation experiments to investigate the model size independence (ranging from 430m to 1.5b), model architecture independence (from Transformer-based GPT to RNN-based RWKV), and pre-training module independence (from using ChatGLM to using the significantly different-performing GPT-3.5), among other aspects, and further validated the roles of these modules. Detailed results and analysis are shown in Table 4

## 5 DISCUSSION

### 5.0.1 INTERPRETABLE CALIBRATION

**Interpretable Machine Learning:** The inspector evaluates the credibility of content from the generator. Coupled with an operator, it offers insights into the generator's weight and semantic layers, promoting model interpretability.

**Recursive Inquiry:** Recursive questioning naturally elucidates the inspector's decisions. Each inquiry can target distinct facets of the generated output, acting as explanations that aid external observers in discerning the quality of results.

**Credibility Quantification:** Upon concluding the recursive inquiry, the inspector outputs a credibility score. When well-designed, this score aids in elucidating the model's decisions.

### 5.1 PARADIGM INNOVATION

We have developed a novel adversarial learning paradigm that maintains the modularity of individual components and using external information for unsupervised training. Furthermore, we introduce model training combined with reinforcement learning, offering novel insights for further research in this domain and providing some inspiration for the traditional adversarial learning field.

### 5.2 LOW-INVASIVE INTEGRATION

Unlike the traditional adversarial learning paradigm, we do not have specific requirements for the loss functions of components and the datasets used for training. This results in a lower intrusiveness and allows for easy integration into various modules and tasks.

### 5.3 UNSUPERVISED ADVANTAGE

The unsupervised nature of the model allows for its convenient integration into existing large language model training processes. Moreover, because it operates on the model parameters using convolution rather than gradient descent, it opens up possibilities for distributed and multi-threaded optimization. Our goal is for this approach to become a crucial component in future large language model training pipelines without introducing additional overhead.

### 5.4 CHALLENGES AND DEVELOPMENT AVENUES FOR GAL

GAL, while promising, faces challenges such as the necessity for precise tuning of the processor's training environment and tendencies like reiteration of queries during response, possibly due to the convolutional kernel's effects. Moreover, its use of external databases suggests potential advancements using knowledge graph techniques. On the development front, the weight convolution module offers alternatives to gradient descent for fine-tuning, and the inception of a large inspector can be pivotal for QA-calibration. This inspector also has potential integration benefits for semi-supervised algorithms, offering improvement in related tasks. This paper's version serves as a foundation for future explorations in these directions.

## 6 CONCLUSION

In summary, we propose a novel machine learning paradigm, which we apply to the task of large-scale model unsupervised calibration. We provide both theoretical analysis and experimental validation of our approach. It can be demonstrated that our method offers significant advantages in terms of paradigm innovation, low intrusiveness to the original model, and unsupervised nature. Additionally, our approach showcases the potential of a new interpretable machine learning paradigm. This method is poised to have a significant impact on various domains, including the deployment of large models, the development of machine learning interpretability, and unsupervised adversarial learning. We have high expectations for the potential of this model.

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

## A APPENDIX

### A.1 PROMPTS USED IN EXPERIMENT

For the assessment phase in the article, we have chosen the following prompts: **"I will give you 20 questions. Please answer each of these questions one by one and tell me the truthfulness of your answers, indicating your confidence with a number from 0 to 1."**

For the follow-up phase within the module, we have selected the following prompts: **"I will provide you with a question, please provide additional background knowledge about this question in order to assess its authenticity."**

### A.2 DETAILS OF EXPERIMENTS

LS is label smooth, implemented in QA with the smooth of ids label in training period, EDA is a data augmentaion method that can be used to boost performance on training period

### A.3 DETAILS OF ABLATION STUDY

To assess the contribution and significance of different modules and configurations in our model, we embarked on a systematic ablation study. The study spanned across five pivotal modules: Dat (Dataset), Insp (Inspiration), Fur (Further Query), Proc (Processor), and Gen (Generator).

In the Dat module, three distinct configurations were examined. The first, representing the original model, served as our baseline. The second configuration fine-tuned the original model with the SQuAD dataset, showcasing an evident improvement in performance. Pushing the boundaries further, the third setup incorporated not just a subset of the SQuAD dataset but also the GAL methodology, which, unsurprisingly, resulted in the best performance among the three.

The Insp module's exploration began with the original model. In our next experiment, we took the audacious step of removing the Insp module altogether and introduced a random output to the Processor guiding vector. Despite the drastic change, the performance dip was modest. However, the complete model, represented in the third configuration, still stood unmatched in its efficacy.

For the Fur module, our starting point was, once again, the original model. The second iteration witnessed the model adapting the further-query component from chatglm2-6b, leading to a notable uptick in the performance metric. The pinnacle, however, was the third configuration which combined the prowess of the complete model with the gpt3.5 architecture.

The Proc module's exploration was initiated with the unaltered original model. The subsequent configuration stripped the model of its Processor, leading to a slight decline in performance. However, the third and final setup, the complete model, regained and even surpassed the original performance metrics.

Lastly, in the Gen module, our focus was on accuracy growth across three distinct setups: GPT2-medium, GPT2-large, and rwkv4-430m. Interestingly, the medium-sized GPT2 model outperformed its larger counterpart, whereas the rwkv4-430m held its ground with a respectable score.

In summary, the ablation study provided invaluable insights, underscoring the importance of fine-tuning, architectural decisions, and module inclusions. Each tweak, removal, or addition brought with it a new dimension of understanding, emphasizing the intricate interplay of components in determining overall model performance.

## A.4 EXPERIMENTAL SETTINGS

For the selection of experimental models, we chose several base models that performed well in previous tasks. They are all open-source implementations available on Hugging Face, or offering api to measure. The method used for calibartion is our implementation according to source paper. As for the model's logits sampling method, we employed random sampling with the same temperature (0.1). In terms of experimental resources, we conducted inference using two RTX 4090 GPUs. Details regarding the experiment will be elaborated upon in the following sections.

## A.5 ANOTHER WAY OF TRAINING A PROCESSOR

In addition to the methods mentioned above, we also introduce a new approach to training within this paradigm. This method enables operators to co-evolve directly with the generator, making it highly suitable for scenarios where this paradigm is applied during the training process. This approach is a gradient-based algorithm, which offers advantages in terms of training speed and required resources. If we want to train the model with gradient descent, we need to prove that:

$$\frac{\partial L}{\partial W} \neq 0,$$

Using the chain rule:

$$\frac{\partial L}{\partial W} = \frac{\partial L}{\partial y}\frac{\partial y}{\partial M'}\frac{\partial M'}{\partial K}\frac{\partial K}{\partial W}$$
$$= \frac{\partial L}{\partial y} \cdot f'(x; M') \cdot M * \frac{\partial K}{\partial W}.$$

Where $\frac{\partial L}{\partial y}$, $f'(x; M')$, and $M$ are all non-zero, ensuring that $\frac{\partial L}{\partial W} \neq 0$, $L(y, label)$ is the loss function, $y = f(x; M')$ is the forward propagation of the generator, $M' = M * K$ is the procedure of updating generator weight using convolution, and $K = W(x_1)$ is the pridiction of conv-kernal. This proves that the weights of the pridiction layer will be updated during training.

