# OpenReview forum: "Generalized Adversarial Learning--An Innovative Unsupervised Paradigm In LLM's Calibration"
_ICLR.cc/2024/Conference — ICLR 2024 Conference Withdrawn Submission_

### Official Review · Reviewer_LxAL · 2023-10-31

**Soundness:** 3 good
**Presentation:** 2 fair
**Contribution:** 3 good
**Rating:** 6
**Confidence:** 3

**Summary:**

This paper tackles the domain of LLM calibration. The authors propose a framework that improves the performance of LLMs in an unsupervised manner. Furthermore, they provide both theoretical analysis and experimental validation of the proposed approach.

**Strengths:**

1. The proposed method is unsupervised in nature which makes it more usable when compared to supervised methods of calibration.
2. The authors provide theoretical analysis of the proposed method, which strengthens their case.
3. The experimental results are promising.

**Weaknesses:**

1. The presentation of this paper can be improved a lot. There are many formatting issues, missing text and grammatical errors. The flow of the paper can also be improved significantly.

**Questions:**

1. In the discussion section the authors mention that the since the proposed methods do not have specific requirements for the loss functions of components and the datasets used for training, it is less intrusive. Can you elaborate a bit more on this? How are you quantifying intrusiveness here?

---

### Official Review · Reviewer_F18K · 2023-10-31

**Soundness:** 2 fair
**Presentation:** 1 poor
**Contribution:** 2 fair
**Rating:** 3
**Confidence:** 4

**Summary:**

This paper introduces Generative Adversarial Learning (GAL), a framework for calibrating question answering (QA) systems without traditional supervision, such as using a held out validation set as part of the calibration process. GAL involves a multi-agent game among a generator (the original pretrained QA model), an inspector, and a processor. The inspector evaluates the generator and generates additional questions to ask the generator. Finally, the processor updates the weights of the generator.

**Strengths:**

- The technique appears to be novel.
- The authors perform a reasonably extensive evaluation and include several performance metrics.
- The IGS score appears to be correlated with accuracy, although I would strongly encourage the authors to make this clear much earlier on in the paper (unless I missed this, which is possible).

**Weaknesses:**

- The paper lacks crucial details about how the algorithm operates. In particular, it is not clear to me how the processor model works.
- Claiming that convolutional kernels are interpretable seems suspect (see 4.3). What is the end objective of interpreting this techniques? Does it help in some concrete way to solve a downstream QA problem, where other techniques fail?
- The authors should include an empirical evaluation of the computational costs incurred by the method. In light of this being missing from the paper, the technique does not appear to be lightweight and non-invasive as the authors claim.

**Questions:**

- The authors refer to the method as non-intrusive -- what exactly does this mean, and how does this differ from existing unsupervised calibration techniques?
- It is not clear to me how the processor model, which is parameterized as a convolutional kernal, works as an optimizer or what the rationale is behind this design choice.
- It is not clear what the confidence matrix refers to. Is this a standard object? Please elaborate on this.
- With regard to the generator model: "It does not need any modifications to its structure or loss function, which is still cross-entropy." I'm not following why the generator still uses the cross entropy loss. Is it not the case that its weights are updated using the processor model? Where do gradients come into play? I assume that supervision signal comes from the inspector?

---

### Official Review · Reviewer_9Xja · 2023-11-08

**Soundness:** 2 fair
**Presentation:** 1 poor
**Contribution:** 2 fair
**Rating:** 3
**Confidence:** 3

**Summary:**

Authors study the problem of adversarial learning in the unsupervised setting which is a relevant problem. They propose an inter-connected framework having three components a) generator, b) processor and c) inspector modules.  Finally, a Proximal Policy Optimization based RL loss objective function is proposed. Authors conduct benchmarking w.r.t recently proposed techniques and ablation studies. The generator is a standard pre-trained LLM, the processor is a weighted convolution model and inspector is the recursive bayesian approximator. Although, authors have claims on interpretability in this paper, strong empirical results supporting the same seem missing.

**Strengths:**

1) This paper to the best of my knowledge does present a framework having a generator, processor and inspector modules for adversarial learning. While the individual components don't seem novel, their application together does come across as something useful especially in the unsupervised paradigm.

2) The individual components are technically sound. Authors also benchmark w.r.t recent techniques

**Weaknesses:**

1) Theoretical analysis in main paper seems under developed and not sure how its useful.
2) Paper touches upon several aspects such as interpretability and adversarial learning without exploring them completely. The paper does come across as being very broad in its claims lacking sufficient evidence supporting all claims made.
3) Ablation studies only a pointer is provided to a table without sharing key insights in the section ( I do see the table caption which by itself does not suffice). I believe ablations are very important for such an inter connected framework with multiple components to validate usefulness of individual components and parameter sensitivity. Sizes of generator models considered, etc are also good variations to explore.
4) Another key weakness I feel is that the writing style does not make the technical contributions very clear, for example for an adversarial learning paper a threat model section is definitely needed explaining the goals, knowledge and tools for both defender and adversary. Paper definitely can go through a thorough editing to surface these contributions better.

Minor comment
Paper has several typos and needs proper editing before submission.

**Questions:**

Most of my questions are mentioned in the weakness section above.

---

### Official Review · Reviewer_fsZm · 2023-11-09

**Soundness:** 1 poor
**Presentation:** 1 poor
**Contribution:** 2 fair
**Rating:** 3
**Confidence:** 3

**Summary:**

The paper proposes a novel unsupervised learning paradigm called "Generalized Adversarial Learning"(GAL) to improve the calibration of large-scale question-answering (QA) models. GAL views adversarial learning as a multi-agent game process, and consists of three components: the generator, the processor, and the inspector. The processor is a module that adjusts the model weights using a convolutional kernel prediction method, while the inspector updates a confidence matrix and a guidance vector. The inspector and the processor cooperate to update the generator's (i.e., the pre-trained LLM) weights in an unsupervised manner.

**Strengths:**

1. The experiments are conducted on several well-known and widely-used LLMs.

**Weaknesses:**

1. The motivation and design principle of the proposed method are not well illustrated.
2. The figures of the paper are not clearly presented.
3. The writings could be greatly improved.

**Questions:**

1. The design principle of the GAL model is not clear. The overall model pipeline of GAL is not clearly described. The model structure is not clearly presented. Why weight convolution model is used for processor?
2. The figures of the paper are not clearly presented. Figure 1 does not clearly presented the model structure. Figure 2 does not show the technical design and contribution of the proposed module. Figure 3 is poorly presented.
3. There are many grammar mistakes and typos. Great efforts should made on improving the writings.